# Robustness of quantum chaos and anomalous relaxation in open quantum circuits

**Takato Yoshimura** [1,2] ✉ **& Lucas Sá** [3,4] ✉

Dissipation is a ubiquitous phenomenon that affects the fate of chaotic quantum many-body dynamics. Here, we show that chaos can be robust against dissipation but can also assist and anomalously enhance relaxation. We compute exactly the dissipative form factor of a generic Floquet quantum circuit with arbitrary on-site dissipation modeled by quantum channels and find that, for long enough times, the system always relaxes with two distinctive regimes characterized by the presence or absence of gap-closing. While the system can sustain a robust ramp for a long (but finite) time interval in the gap-closing regime, relaxation is "assisted" by quantum chaos in the regime where the gap remains nonzero. In the latter regime, we prove that, if the thermodynamic limit is taken first, the gap does not close even in the dissipationless limit. We complement our analytical findings with numerical results for quantum qubit circuits.

Quantum chaos is a powerful statistical framework for studying generic complex many-body quantum systems in and out of equilibrium. Relating the universal late-time dynamics and typical observables of such systems to statistical properties of random matrices allows for a better theoretical understanding of issues ranging from thermalization in statistical mechanics[1,2], to information scrambling in quantum information science[3–5], and the holographic principle in quantum gravity[6–9]. In closed quantum systems with unitary dynamics, the spectral form factor (SFF)[9–14] assumes a particularly crucial role, as it establishes a concrete connection between dynamics and spectral correlations[15–21]. In this context, the repulsion of eigenvalues and spectral rigidity of the spectra of ergodic systems manifest as the presence, at late times, of a ramp in the SFF, one of the hallmark signatures of quantum chaos.

Random quantum circuits (RQCs)[22–32] provide an excellent platform to investigate the out-of-equilibrium dynamics and quantum chaos features of local strongly interacting quantum systems, due to their analytical tractability (sometimes only in the limit of large Hilbert space dimension $q$[24,25]), simple numerical implementation, and simulability in emergent quantum computing platforms[33]. In particular, the emergence of the ramp and deviations from random-matrix universality can be analytically understood in a particular family of Floquet RQCs, the random phase model (RPM)[25,34–36], for which the exact SFF[25], autocorrelation functions[36], and the out-of-time-ordered correlator (OTOC)[36] were computed at $q \to \infty$.

By now, much is known about unitary circuits and pure-state dynamics, even in the presence of external measurements[32,37–43]. Nevertheless, present-day quantum computers are noisy, which makes it important to study nonunitary dissipative circuits, implemented by lossy or imperfect hardware, and modeled by quantum-channel gates (the dissipative generalization of unitary gates)[44–53]. More broadly, all physical systems are, to some degree, influenced by interactions with an environment or errors in controlling protocols. It is therefore of great importance to test the robustness of quantum chaotic features in the presence of dissipation and quantify the dissipative corrections to the quantities most commonly used to characterize chaos, e.g., the ramp of the SFF, the butterfly velocity[22,23,54], or the quantum Lyapunov exponent[8,55].

Previous research in similar directions has focused on the entanglement dynamics and information scrambling in brickwork open

[1]All Souls College, Oxford, UK. [2]Rudolf Peierls Centre for Theoretical Physics, University of Oxford, Oxford, UK. [3]TCM Group, Cavendish Laboratory, University of Cambridge, JJ Thomson Avenue, Cambridge, UK. [4]CeFEMA, Instituto Superior Técnico, Universidade de Lisboa, Av. Rovisco Pais, Lisboa, Portugal. ✉e-mail: takato.yoshimura@physics.ox.ac.uk; ld710@cam.ac.uk

RQCs[46–51], but the spectral correlations and relaxation dynamics of dissipative Floquet circuits have remained unexplored. Distinct open-system extensions of the SFF have been proposed that capture different aspects of dissipative quantum chaos[56–70]. In particular, the dissipative form factor (DFF)[56] extends the dynamical definition of the SFF and is given by the trace of the quantum channel. The late-time behavior of the DFF is controlled by the spectral gap and was used to compute it for random Liouvillians[56], while the finite-time behavior captures dynamical phase transitions[57]. It does not, however, measure the correlations of eigenvalues in the complex plane, which are captured, instead, by a different quantity, the dissipative spectral form factor[59–64].

This motivates us to ask a fundamental question: What, if any, are the universal features of the DFF in quantum chaotic systems? To take a step forward in answering this question, we introduce a minimally-structured many-body chaotic open Floquet system using a dissipative extension of the RPM, which we call the dissipative random phase model (DRPM). To test the robustness of the chaotic dynamics against the presence of dissipation, we analytically compute its DFF for arbitrary on-site dissipation in the limit of large local Hilbert space dimension ($q \to \infty$). We find that, when dissipation is weak enough, the ramp, which signals the presence of level repulsion and spectral rigidity, persists over a timescale that is parametrically larger than the Thouless time, before the DFF finally decays exponentially at a rate set by the spectral gap. Remarkably, the gap does not necessarily close in the limit of vanishing dissipation if the thermodynamic limit is taken first, a phenomenon recently observed in several dissipative many-body systems[71–74] and dubbed anomalous relaxation[72]. We expect these features of the DFF seen in the DRPM to be generic in open Floquet many-body systems without conservation laws. To further support this claim, we provide numerical evidence by simulating a brickwork Floquet circuit acting on qubits ($q = 2$).

## Results

### Exact DFF of the DRPM

The dynamics of the DRPM are generated by the Floquet operator of the RPM and on-site dissipation prescribed by quantum channels. Let us start by recalling the definition of the RPM[25], see also Fig. 1b. We consider a spin chain of $L$ sites where the on-site Hilbert space dimension is $q \in \mathbb{Z}^+$. The time-evolution of the model is discrete and governed by the Floquet operator $W(t) = W^t$ where the operator $W$ is made of two layers, $W = W_2 W_1$. First, $W_1 = U_1 \otimes \cdots \otimes U_L$ consists of $q \times q$ on-site random unitaries $U_x$ that are Haar distributed. Second, $W_2$ induces interactions among adjacent sites and acts on the basis state $|a_1\rangle \otimes \cdots \otimes |a_L\rangle \in (\mathbb{C}^q)^L = \mathcal{H}$ diagonally with the phase $\exp(i\sum_{x=0}^{L} \varphi_{a_x, a_{x+1}})$. Each $\varphi_{a_x, a_{x+1}}$ is independently Gaussian distributed with mean zero and variance $\varepsilon/2 > 0$.

In the DRPM, each Floquet step is followed by the action of the quantum channel $\Phi$ that describes local dissipation. In this paper, we consider quantum channels that factorize into a product of decoupled single-site channels, i.e., $\Phi = \Phi_1 \otimes \cdots \otimes \Phi_L$. The local quantum channel $\Phi_x$ acts on the state $\rho_x$ at site $x$ as $\Phi_x(\rho_x) = \sum_{i=0}^{k-1} M_i \rho_x M_i^\dagger$, where $k$ is the number of channels and $M_i$ are Kraus operators that satisfy $\sum_{i=0}^{k-1} M_i^\dagger M_i = I$, with $I$ the identity (the channels at different sites $x = 1, \ldots, L$ can be different). We normalize the Kraus operators as $q^{-1}\text{Tr}(M_i M_j^\dagger) = \eta_i \delta_{ij}$ for some $0 \le \eta_i \le 1$ such that $\sum_{i=0}^{k-1} \eta_i = 1$ (which follows from complete positivity and trace-preservation of the channel). A complete time step of the nonunitary Floquet circuit is best represented as an operator on the doubled Hilbert space $\mathcal{H} \otimes \mathcal{H}^*$, which acts on the vectorized density operator as a matrix $\mathcal{W} = \Phi(W \otimes W^*)$. The entire circuit at time $t$ is given by $\mathcal{W}^t$.

To understand the effect of dissipation on quantum chaos, the central object in this paper is the DFF, originally introduced[56] for Lindbladian dynamics as $F(t) = \text{Tr}\, e^{t\mathcal{L}}$, where the Lindbladian $\mathcal{L}$ acts on the doubled Hilbert space. The DFF reduces to the standard SFF, $F(t) = |\text{Tr}\, e^{-iHt}|^2$, where $H$ is the Hamiltonian, in the absence of dissipation. To generalize it to Floquet dynamics, we simply replace $e^{t\mathcal{L}}$ by the quantum channel $\mathcal{W}^t$ of our circuit[34,70,75]. Writing $\mathcal{W}^t$ in the vectorized Kraus representation, $\mathcal{W}^t = \sum_j K_j \otimes K_j^*$, with global Kraus operators $K_j = \prod_{\tau=1}^{t}[(\prod_{x=1}^{L} M_{j_{\tau x}})W]$ and $M_{j_{\tau x}}$ the local Kraus operator at site $x$ and time step $\tau$, the DFF for the Floquet quantum circuit can be expressed as[75]

$$F(t) = \text{Tr}\, \mathcal{W}^t = \sum_j |\text{Tr} K_j|^2. \tag{1}$$

Figure 1a depicts the diagrammatic representation of $F(2)$ with the system size $L = 3$. Interpreting $M_{j_{\tau x}}$ as a generalized measurement, the quantity $|\text{Tr} K_j|^2$ is the SFF of a monitored quantum circuit[37–39], where, after each time step and for each site, a measurement is performed and the outcome recorded. We thus see that the DFF is the equal-weight average over all such possible measurement histories, as relevant for an unmonitored quantum circuit (e.g., in the presence of an external environment or noise source), where one does not keep track of the measurement outcomes.

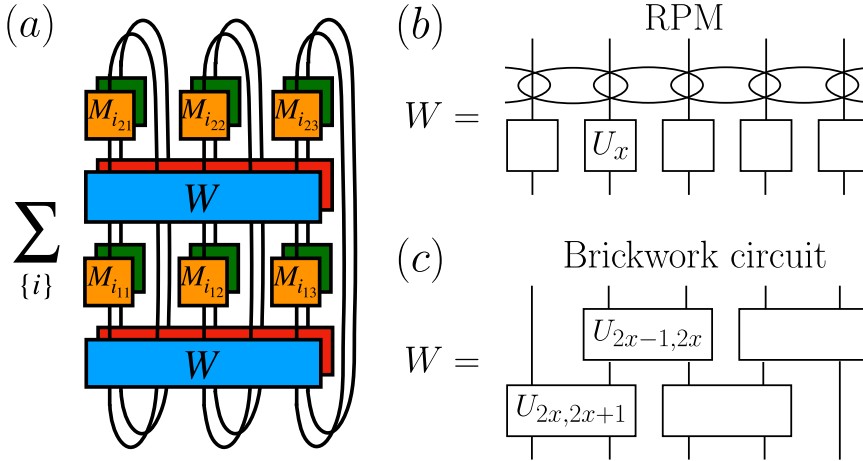

**Fig. 1 | Schematic representation of the computations in this work. a** DFF for a nonunitary open quantum circuit for $t = 2$ and $L = 3$. Lines represent spins. Blue and red gates represent the unitary Floquet operator $W$ and its conjugate $W^*$. Yellow and green on-site gates are Kraus operators $M_{i_{x\tau}}$ and their conjugates $M_{i_{x\tau}}^*$. **b** and **c** Circuit architecture of unitary contribution $W$ to the DFF. **b**: RPM; squares denote independently drawn single-site Haar unitaries, while ellipses denote the diagonal random phase couplings between adjacent sites. **c**: brickwork circuit; the rectangles denote independently drawn two-site Haar random unitaries.

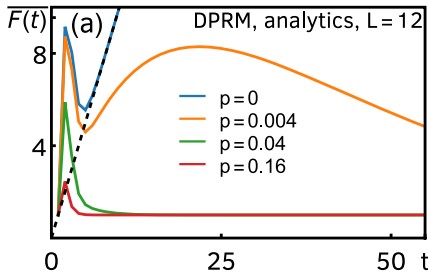

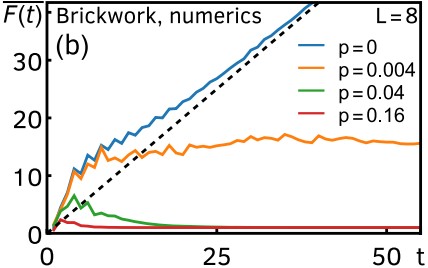

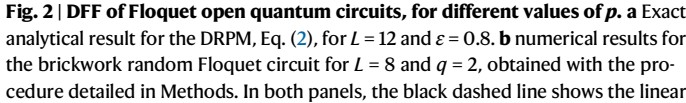

**Fig. 2 | DFF of Floquet open quantum circuits, for different values of $p$. a** Exact analytical result for the DRPM, Eq. (2), for $L = 12$ and $\varepsilon = 0.8$. **b** numerical results for the brickwork random Floquet circuit for $L = 8$ and $q = 2$, obtained with the procedure detailed in Methods. In both panels, the black dashed line shows the linear ramp as predicted by RMT. In both panels, the DFF is above this line for small $t$ because of the existence of the Thouless peak, while for large $t$ and $p > 0$, it decays to a value below the dashed line.

In what follows, we evaluate the ensemble average of the DFF, Eq. (1), which we denote by $\overline{F(t)}$, at large $q$, where Haar-averaging becomes substantially simplified, for any single-site quantum channels. We stress that, in the RPM, the SFF retains its main features even in the large-$q$ limit[31]. This is because the mechanism that causes the early-time deviation of the SFF from the RMT behavior at finite $q$ is still kept intact at large $q$ in the RPM[34]. We thus expect that the large-$q$ DFF of the DRPM also captures the essential features of the DFF of generic open Floquet systems without conservation laws at finite $q$.

We follow the same strategy as in the computation of the SFF of the RPM[25], which is to calculate the DRPM diagrammatically; see Methods for the details of the computation. Haar-averaging at each site induces local pairings of indices labeling on-site unitaries $U_x$ and $U_x^*$, but it is straightforward to show that only those corresponding to cyclic pairings of indices contribute at $q \to \infty$, which is also the case in the SFF (in the SFF, these $t$ pairings give rise to the late-time ramp $\sim t$)[25]. We label each pairing by $s = 0, ..., t - 1$ ($s = 0$ corresponds to the pairing where both indices of $U_x$ are contracted with those of $U_x^*$ on the same time slice), and it turns out that diagrams associated with any $s \neq 0$ pairing carry the same factor $\kappa := \sum_i \eta_i^t$, while diagrams with an $s = 0$ pairing simply come with the factor 1 (see Methods). Since different pairings on two neighboring sites—a configuration which we call a domain wall—induce the statistical cost $e^{-\varepsilon t}$ upon averaging with respect to the phase $\varphi$, similarly to the SFF of the RPM[25], we find that the DFF of the DRPM at large $q$ can be succinctly expressed as[75]

$$\overline{F(t)} = \text{Tr}\,\hat{T}^L, \quad \hat{T} = TD, \tag{2}$$

where $T = (1 - e^{-\varepsilon t})I + e^{-\varepsilon t}E$ ($E$ is the matrix completely filled with ones) is the $t \times t$ transfer matrix of the SFF acting on the $t$ pairing degrees of freedom and $D = \kappa I + (1 - \kappa)|0\rangle\langle 0|$ is the on-site dissipative contribution. Here, $|s\rangle$ denotes the vector with a 1 in the $s$th coordinate and 0 elsewhere, thus $|s\rangle\langle s|$ is the projection matrix to the pairing labeled by $s$. The trace follows from periodic boundary conditions. In writing Eq. (2), we have assumed that the same quantum channel acts at each site; if this condition is relaxed, the DFF reads as $\overline{F(t)} = \text{Tr}\prod_x (TD_x)$.

In ref. 25, it was shown that the SFF of the RPM [corresponding to Eq. (2) at $\kappa = 1$: $\overline{F(t)}|_{\kappa=1} = \text{Tr}\,T^L$] displays a ramp for times larger than the Thouless time $t_{\text{Th}} = \varepsilon^{-1}\log L$. We are interested in how this behavior is modified by dissipation, which is quantified by the DFF. We emphasize that Eq. (2) is valid for any choice of local dissipative gate and all information on the latter is encoded in the weight $\kappa$.

**Relaxation in the DRPM**

While the exact large-$q$ DFF Eq. (2) is valid for any choice of quantum channels, to make the analysis concrete, we shall focus on a quantum channel where all $\Phi_x$ are the same fixed depolarizing channel, $\Phi_x(\rho_x) = (1 - p)\rho_x + pI/q$, when studying the behavior of the DFF either analytically or numerically. The corresponding Kraus operators are

given by $M_0 = \sqrt{1 - p(q^2 - 1)/q^2}I$ and $M_i = \sqrt{p/q^2}P_i$ ($i = 1, ..., q^2 - 1$), where $0 \leq p \leq 1$ is the probability that the spin is depolarized and $P_i$ are Hermitian operators that form a traceless orthonormal basis of on-site operators with $q^{-1}\text{Tr}(P_iP_j) = \delta_{ij}$. The normalization factor $\eta_i$ for this channel is $\eta_0 = 1 - p(q^2 - 1)/q^2 \approx 1 - p$ (where the approximate expression holds in the limit $q \to \infty$) and $\eta_i = p/q^2$. We thus have $\kappa = (1 - p)^t$ in the large-$q$ limit.

The main features of the DFF of the DRPM are shown in Fig. 2a. We first notice that, after a sharp drop from the initial-time peak $\overline{F(0)} = q^{2L}$ [not visible in the scale of Fig. 2a] to $\overline{F(1)} = 1$, the DFF develops a second peak, which we call the Thouless peak. After this, there is a linear ramp for $p = 0$ (closed system), which is still sustained for small $p$. Dissipation in the DRPM induces relaxation to its unique steady state, which is seen in the late-time exponential decay of the DFF to a constant plateau of value one. For larger $p$, the DFF decays immediately after the Thouless peak, and there is no ramp.

As we argued earlier, we expect that the DRPM at large $q$ embodies the main structures in open Floquet systems without conservation laws, and therefore that the above behavior of the DFF is universal in these systems. To corroborate this claim, before moving to the detailed analysis of the DRPM, let us provide independent numerical support. We numerically simulated a qubit ($q = 2$) random Floquet circuit with brickwork structure, see Fig. 1c. Each unitary Floquet step has now two layers, where $4 \times 4$ Haar-random unitaries couple sites $2x - 1$ and $2x$ ($x = 0, ..., \lfloor L/2 \rfloor$) in the first half timestep, and sites $2x$ and $2x + 1$ in the second. We take open boundary conditions to allow us to consider odd $L$ and to render the Thouless peak more visible[34]. The local dissipative channels are chosen as translationally-invariant depolarizing channels as before. In Fig. 2b, we numerically computed the DFF for this circuit, see Methods. Although the circuit architecture is different and $q$ is finite, we still observe the same qualitative behavior of the DFF as in the DRPM (for this $q$ and $L$ the Thouless peak is less prominent[34]).

All of the above characteristics can be analytically understood for the DRPM using the exact result of Eq. (2). The Thouless peak [the peak at small times, see the green region in Fig. 3a] is a consequence of each site behaving independently at early times. This means that the early-time behavior of the DFF can be understood from a collection of single-site systems subject to dissipation. The DFF of the uncoupled single site (i.e., $\varepsilon = 0$, $L = 1$) dissipated by the depolarizing channel behaves as $1 + (t - 1)\kappa$ at large $q$, thus the DFF of the DRPM at early times grows as $\overline{F(t)} \sim [1 + (t - 1)\kappa]^L$.

At late times, the effect of the coupling between different sites sets in, inducing the decay of the Thouless peak and the relaxation of the DFF. The late-time physics after the Thouless peak can be accurately approximated by the pairing-domain-wall expansion (see Methods),

$$\overline{F(t)} \simeq 1 + t\kappa^L + tL\kappa e^{-2\varepsilon t} + \frac{t^2L^2}{4}\kappa^L e^{-2\varepsilon t} + \cdots, \tag{3}$$

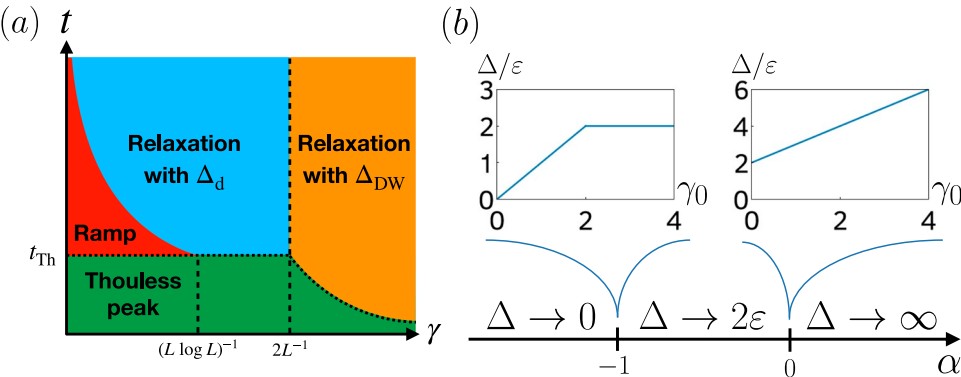

**Fig. 3 | Relaxation in the DFF of the DRPM for large $L$. a** Different dynamical regimes in the DRPM as a function of $\gamma = -\varepsilon^{-1}\log(1-p)$. The boundary between the red and blue regions is given by $t_d$, while the dotted line separating the green region from the remaining ones corresponds to $t_{DW}$. **b** Asymptotic $L \to \infty$ behavior of the gap for different $\alpha$ in $\gamma = \gamma_0 L^\alpha$. It is independent of $\gamma_0$, except at the special points $\alpha = -1$ and $\alpha = 0$.

where we only kept up to two domain walls, as they decay most slowly and hence capture the tail of the Thouless peak. The ellipsis refers to the terms that contain more domain walls. It should be stressed that the late-time expansion Eq. (3) is still applicable to any quantum channel.

In what follows, we focus on depolarizing channels for concreteness. It is convenient to introduce the effective dissipation strength $\gamma = -\varepsilon^{-1}\log(1-p) > 0$, with which the expansion Eq. (3) can be written as

$$\overline{F(t)} = 1 + F_d(t) + F_{DW}(t) + \cdots . \tag{4}$$

Here,

$$F_d(t) = t e^{-\gamma L \varepsilon t} \tag{5}$$

is the direct contribution from dissipation and shows an initial linear ramp followed by an exponential decay at a rate $\Delta_d = \varepsilon\gamma L$. We call the peak associated with $F_d(t)$ the dissipation peak, see e.g., the last peak of the orange curve in Fig. 2. Likewise,

$$F_{DW}(t) = tL e^{-\varepsilon t(2+\gamma)} + \frac{t^2 L^2}{4} e^{-\varepsilon t(2+\gamma L)} \tag{6}$$

describes the behavior stemming from domain-wall physics. Note that the first term in $F_{DW}(t)$ becomes larger than the second term when $t \gg \log L/(\varepsilon\gamma L) =: t_*$. The decay of the peak in $F_{DW}(t)$, i.e., the Thouless peak, is governed by the asymptotic rate $\Delta_{DW} = \varepsilon(2+\gamma)$. The overall asymptotic decay of the DFF is set by the slower of the decay rates (the gap):

$$\Delta = \min\{\Delta_d, \Delta_{DW}\} = \min\{\varepsilon\gamma L, \varepsilon(2+\gamma)\}. \tag{7}$$

Below, we demonstrate how the competition between the decay of the Thouless and dissipation peaks amounts to a rich relaxation behavior. Since the relaxation timescales of the DRPM depend on the scaling of

**Table 1 | Definition and system-size dependence of the various timescales arising in the relaxation dynamics of the DRPM**

| |
|---|
| $t_* = \log L/(\varepsilon\gamma L) \sim L^{-\alpha-1}\log L$ |
| $t_{DW} = 2\varepsilon^{-1}\log L/(2+\gamma L) \sim (2+\gamma_0 L^{\alpha+1})^{-1}\log L$ |
| $t_{Th} = \varepsilon^{-1}\log L$ |
| $t_d = 1/(\varepsilon\gamma L) \sim L^{-\alpha-1}$ |

the parameter $\gamma$ with $L$, we parameterize $\gamma = \gamma_0 L^\alpha$, where $\alpha \in \mathbb{R}$ and $\gamma_0$ is independent of $L$. Often, a local dissipation strength that is independent of the spatial extent of the system (i.e., $\alpha = 0$) is the most natural choice. However, we note that a rescaling of the local dissipation strength with system size (in particular, with $\alpha < 0$) is in some cases necessary to get a well-defined theory in the thermodynamic limit[71]. In the following, we analyze in more detail the behavior of the DFF for different $\alpha$, taking $L \to \infty$. Since it is necessary to introduce multiple timescales with different system-size dependence, to avoid confusion, they are summarized in Table 1.

**Robustness of the ramp**

We start with the case $\alpha < -1$, i.e., $\gamma$ sufficiently small. In this regime, $t_*$ diverges in the thermodynamic limit and, therefore, $F_{DW}(t)$ is always dominated by its second term at finite times. The Thouless peak [green region in Fig. 3a] thus decays over the timescale $t_{DW} = 2\log L/[\varepsilon(2+\gamma L)] \simeq \varepsilon^{-1}\log L = t_{Th}$ and is well-separated from, and occurs at a much earlier time than, the dissipation peak. In particular, the build-up of the dissipation peak can be thought of as a remnant of the ramp in the SFF, and the width of the peak can be arbitrarily stretched as we take $\gamma_0 \to 0$ [red region in Fig. 3a]. Eventually, after a time $t_d = \Delta_d^{-1}$, the dissipation peak starts to decay at a rate $\Delta_d$ [blue area in Fig. 3a], where the gap $\Delta = \Delta_d = \varepsilon\gamma_0 L^{\alpha+1}$ closes as $L \to \infty$, see Fig. 3b. The timescale $t_d$ up to which the ramp survives is parametrically larger than the Thouless time (and, in particular, diverges in the thermodynamic limit), showing that the ramp of the closed RPM, which signals the chaotic correlations of the system, is robust against the addition of small amounts of dissipation to the system.

The robustness of the ramp persists for any $\alpha < -1$, but is destroyed by logarithmic corrections in the limit $\alpha \to -1^-$. Indeed, when the timescales of the Thouless and dissipation peaks become comparable, $t_{DW} \sim t_d$, i.e., $\gamma \sim 1/(L\log L) =: \gamma_{ramp}$, the ramp ceases to exist. Thus, when $\gamma \gtrsim \gamma_{ramp}$ (but still $\alpha < -1$), the two peaks have an overlap, and as a result the DFF shows a two-stage relaxation that is divided by the timescale $t_{DW}$: when $t \lesssim t_{DW}$ the exponent of the (exponential) decay is given by $\Delta_{DW} = 2\varepsilon$, whereas when $t \gtrsim t_{DW}$ the decay is dictated by the gap $\Delta_d$. We depict the two-stage relaxation for a large $L = 100$ in Fig. 4. This is the intermediate regime where dissipation is strong enough to suppress the ramp completely but not enough to overwhelm the domain-wall contribution.

**Anomalous relaxation**

As we increase the value of $\alpha$, the system enters another scaling regime with $-1 \leq \alpha \leq 0$. In this case, the gap $\Delta$ does not close and remains a positive finite constant as $L \to \infty$ [see Fig. 3a]. Because $t_*$ now goes to zero in the thermodynamic limit, the first term in $F_{DW}(t)$ is always

dominant after $t_{DW} \approx t_{Th}/(2+\gamma)$ (the dotted line separating the green and yellow regions in Fig. 3a ceases to be constant). The most dramatic change, however, is that the gap, Eq. (7), changes as the dissipation peak is completely engulfed by the Thouless peak, which happens when $\Delta_d = \Delta_{DW}$, i.e., $\gamma = 2/L$ [corresponding to the dashed line separating the blue and yellow regions in Fig. 3b]. That is, at $\alpha = -1$, as a function of $\gamma_0$, the gap changes from $\Delta = \Delta_d = \varepsilon\gamma_0$ to $\Delta = \Delta_{DW} = \varepsilon(2+\gamma_0/L)$ [see the left inset of Fig. 3b]. Interestingly, from that point on, and also for all $-1 < \alpha < 0$, the gap is effectively independent of the dissipation strength $\gamma_0$ at large $L$, $\Delta = \Delta_{DW} = 2\varepsilon$. For $\alpha = 0$, it depends again on $\gamma_0$, increasing linearly from its initial value, $\Delta = \varepsilon(2+\gamma_0)$ [see the right inset of Fig. 3b]. Finally, when $\alpha > 0$, the effect of the dissipation peak becomes completely negligible and the decay is solely controlled by the Thouless peak. The gap $\Delta_{DW} = \varepsilon(2+\gamma_0 L^\alpha) \approx \varepsilon\gamma_0 L^\alpha$ diverges in the thermodynamic limit [see Fig. 3b].

From the previous discussion, it follows that if $-1 < \alpha \le 0$, the gap does not vanish (as one would naively expect) and stays finite in the dissipationless limit $\gamma_0 \to 0$, provided that the thermodynamic limit $L \to \infty$ is taken first. This remarkable non-commutativity of limits implies that the system relaxes at a finite rate even in the absence of an explicit coupling to the bath. In particular, the gap is independent of $\gamma_0$, and the relaxation is, therefore, driven not by the dissipation but instead by domain walls, which are one of the major consequences of the interplay between locality and the underlying RMT structure in Floquet many-body systems without conservation laws[25,34]. A similar chaos-driven relaxation was first observed[71,72] in the weak-dissipation regime dissipative Sachdev-Ye-Kitaev model[57,71,72,76,77] and termed anomalous relaxation[72]. It was conjectured to be a generic feature of open chaotic quantum systems and has been, since then, observed in other strongly interacting dissipative systems[73,74].

To further support the ubiquity of anomalous relaxation, we also computed the spectral gap for the brickwork circuit at $q = 2$ using a

power iteration method (see Methods), depicted in Fig. 5a for different $p$ and $L$ up to 9. While, strictly speaking, anomalous relaxation requires taking the thermodynamic limit, we argue that the numerical results at finite $L$ support its presence. The gap increases with $L$ for small $p$, while collapsing to an $L$-independent single value for $p \gtrsim 0.12$. The extrapolation to $L \to \infty$ gives approximately a constant value for a moderate range of $p$. We cannot extend the extrapolation to values of $p \lesssim 0.07$ because this would amount to taking the small-$p$ limit first. Moreover, the slope near $p = 0$ is increasing with increasing $L$, signaling that, indeed, the regime of validity of the extrapolation is increasing. In the limit $L \to \infty$, we expect this slope to diverge, leading to an extension of the constant extrapolation value down to $p = 0$. These two features—a linear growth of the gap in a window of $p$ that decreases with $L$ and a convergence to an (almost) constant—are the smoking-gun signatures of anomalous relaxation and they are also present, for example, in the exact finite-$L$ expression of the gap of the DRPM. To see this, we fitted an exponential decay to the late-time regime of the exact transfer-matrix expression of the DFF of the DRPM, Eq. (2), see Fig. 5b. The resulting gap (i.e., decay rate) as a function of $p$ shows the same qualitative features as discussed above. Incidentally, this also provides a check on the accuracy of the domain-wall expansion, as we find an excellent agreement between the fit to Eq. (2) and the domain-wall expression, Eq. (7).

## Purity dynamics

Having seen that the DFF unveils a rich relaxation mechanism in the DRPM caused by the competition between quantum chaos and dissipation, it is natural to ask what kind of signatures of dissipative quantum chaos could be displayed in other quantities. Motivated by this, we look into the (ensemble-averaged) purity $\overline{\mathcal{P}_A(t)}$. Let $A = \{1, ..., L_A\}$ be the left part of the system and trace out its complement $\bar{A}$, yielding the reduced density matrix $\rho_A(t) = \mathrm{Tr}_{\bar{A}}(\mathcal{W}^t[|\psi\rangle\langle\psi|])$ where $|\psi\rangle$ is the initial product state. The bipartite purity is then defined as

$$\mathcal{P}_A(t) = e^{-S_A^{(2)}} = \mathrm{Tr}_A[\rho_A(t)^2], \tag{8}$$

where $S_A^{(n)}$ is the $n$th Rényi entropy, $S_A^{(n)} = \log(\mathrm{Tr}[\rho_A(t)^n])/(1-n)$. Without dissipation, $S_A^{(2)}$ in general grows linearly in time, causing the exponential decay of the bipartite purity. Again with the aid of diagrammatic techniques, we compute the ensemble-averaged bipartite purity $\overline{\mathcal{P}_A(t)}$ at large $q$ (see Methods for the derivation). For example, we obtain the following simple expression in the case of depolarizing channels:

$$\overline{\mathcal{P}_A(t)} = e^{-2(1+\gamma L_A)\varepsilon t}. \tag{9}$$

We contrast this result with the previous computation of the DFF. There is no scaling of $\gamma$ and $L_A$ with the system size such that the dissipative correction to the bipartite purity gap $2\varepsilon\gamma L_A$ becomes a finite

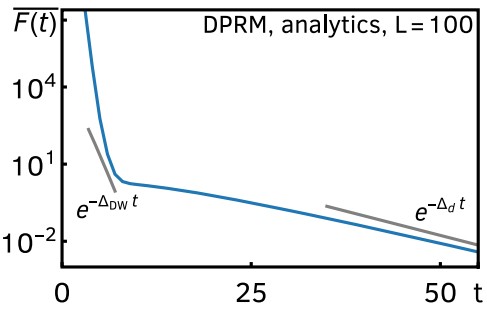

**Fig. 4 | Two-stage relaxation of the exact DFF for the DRPM.** We evaluate Eq. (2) for $L = 100$, $\varepsilon = 0.8$, and $p = 1 - e^{-\varepsilon/L \log L} \approx 0.0017$. It shows a two-stage exponential relaxation with an intermediate-time exponent $\Delta_{DW} \approx 1.8$ and a late-time exponent $\Delta_d \approx 0.17$, as predicted by the domain-wall expansion.

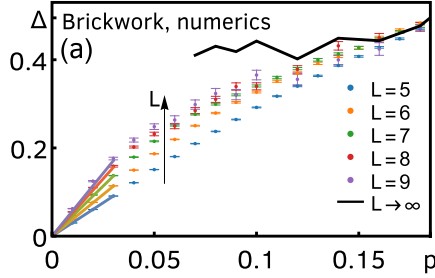

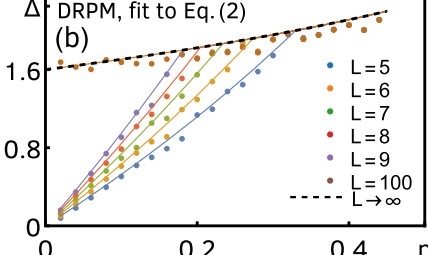

**Fig. 5 | The spectral gap of Floquet open quantum circuits as a function of $p$ for different $L$. a** Numerical results for the brickwork circuit at $q = 2$. The colored dots give the average spectral gap obtained from power iteration (see Methods), while the black line gives an extrapolation to $L \to \infty$ (linear in $1/L$). The colored full lines give the linear slope of the gap at small $p$ for the respective values of $L$. **b** Comparison of the exact and domain-wall expansion results (both analytical) for the DRPM with $\varepsilon = 0.8$. The colored dots are obtained from an exponential fit to the late-time decay of the exact transfer-matrix expression of the DFF, Eq. (2), and they are compared to the domain-wall-expansion result (colored full lines), Eq. (7). The dashed line is the analytical $L \to \infty$ gap.

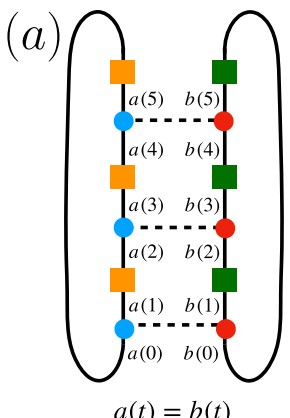 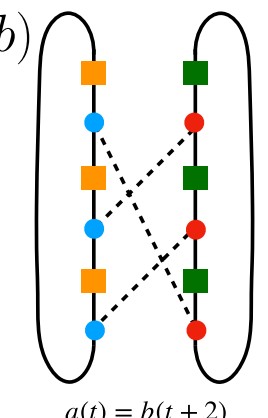 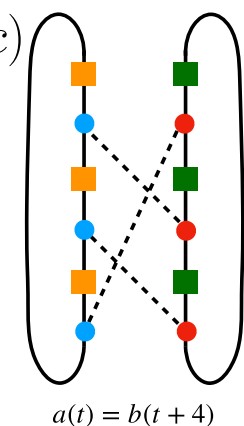

$$a(t) = b(t) \qquad a(t) = b(t+2) \qquad a(t) = b(t+4)$$

**Fig. 6 | Diagrammatic representation of the leading contributions to the DFF as $q \to \infty$.** Blue and red circles represent $U_n$ and $U_n^*$, while yellow and green squares denote the Kraus operators $M_j$ and $M_j^*$. The dashed lines denote contractions of unitaries. $a(\tau)$ and $b(\tau)$ denote the indices of the unitaries $U_n$ and $U_n^*$, respectively (e.g., the first unitary has $U_n$ has indices $[U_n]_{a(0),a(1)}$). The three diagrams depict the leading cyclic pairings at $t = 3$, corresponding to **a** $a(t) = b(t)$, **b** $a(t) = b(t+2)$, and **c** $a(t) = b(t+4)$.

constant when the thermodynamic limit is taken before the weak-dissipation limit. Anomalous relaxation—understood as the noncom-mutativity of these two limits, with the appropriate order yielding a finite nonzero constant—is therefore absent in the ensemble-averaged purity $\overline{\mathcal{P}_A(t)}$ (this being said, for the moment we cannot rule out the possibility that the quenched average $\exp\{-\overline{S_A^{(2)}(t)}\}$ exhibits anomalous relaxation). We thus draw the important lesson that dissipative quantum chaos does not leave the same imprint in the relaxation of all quantities, and a judicious choice of observable is necessary.

Interestingly, there are situations where anomalous relaxation happens at the subleading order in $q^{-1}$ and is, thereby, masked in the $q \to \infty$ limit. A simple example is the average of the standard purity $\mathcal{P}(t) = \mathrm{Tr}[\rho(t)^2]$ where $\rho(t) = \mathcal{W}^t[|\psi\rangle\langle\psi|]$. A similar computation as in the bipartite purity shows that the averaged purity at $q \to \infty$ reads $\overline{\mathcal{P}(t)} = (1-p)^{Lt}$. Analogously to the previous case, there is no anomalous contribution to the purity gap. This is so because anomalous relaxation requires domain walls, which are absent at the leading order in $q^{-1}$ in the averaged purity but contribute at higher order. Therefore, we expect that the averaged purity exhibits anomalous relaxation at subleading order in $q^{-1}$.

## Discussion

We have introduced the dissipative random phase model and studied its dissipative form factor. We found that the interplay of dissipation and quantum chaos gives rise to intriguing behaviors during relaxation that change dramatically as the scaling of the dissipation strength in the system size varies. We also observed that one signature of quantum chaos in the spectral form factor, i.e., the late-time ramp, is robust when dissipation is sufficiently weak, $\gamma \lesssim (L \log L)^{-1}$, and its remnant persists over a timescale controlled by the gap. We also showed that the gap does not necessarily close in the thermodynamic limit even when the dissipation strength is set to zero. We expect these behaviors to be generic among open quantum many-body systems without conservation laws, and to further support this claim, we provided compelling numerical evidence by simulating another prototypical open Floquet circuit. Since quantum-channel circuits are realistic models of noisy intermediate-scale quantum (NISQ) computers and our results are largely independent of the precise choice of channel, it would be interesting to look for these signatures of dissipative quantum chaos experimentally in NISQ devices, using, e.g., an extension of the protocol proposed in ref. 78.

Strikingly, our work reveals the underlying mechanism of anomalous relaxation in dissipative quantum chaotic systems without conservation laws. As we have seen, anomalous relaxation occurs when relaxation is controlled by the Thouless peak in the DRPM. This implies that anomalous relaxation is essentially caused by domain-wall physics, which is neither an artifact of the large-$q$ limit nor the peculiarity of the RPM. Indeed, it was argued in ref. 34 that the domain-wall physics is universal in Floquet many-body systems without conservation laws even at finite $q$ where effective domain walls rule the deviation from RMT behavior in the SFF. This observation further suggests that the presence of conservation laws would spoil anomalous relaxation, as the Thouless peak no longer controls the onset of RMT behavior in the SFF[30]. This is also consistent with the recent study that pointed out the importance of having no conservation laws in the system for anomalous relaxation to take place[74]. We will report a systematic study of the role of conservation laws in anomalous relaxation in forthcoming work.

## Methods

### Diagrammatic computation of the DFF

**Haar averaging in the presence of dissipation.** The computation of the DFF proceeds similarly to that of the standard SFF[25]. We first Haar-average each on-site diagram, which induces local pairings among on-site unitaries. We then enumerate the weight of every contributing local pairing at large $q$ and analyze what phase contribution we get upon averaging with respect to the phase $\varphi$ for every possible pair of two pairings on two neighboring sites. This gives rise to the transfer matrix $T$ with the on-site matrix $D$ of the DFF, allowing us to compute the DFF as $\overline{F(t)} = \mathrm{Tr}(TD)^L$.

Let us start with the Haar average at site $n = 1, ..., L$. It is useful to represent on-site diagrams as in Fig. 6, where we depict the case of $t = 3$ [it corresponds to a rotation by 90° around a vertical axis of the diagram of Fig. 1a] and time runs upward. Squares represent the on-site quantum channels $M_{j_{x\tau}}$ and $M_{j_{x\tau}}^*$, whereas circles are the Haar unitaries and their conjugates. Haar averaging then induces $(t!)^2$ different pairings of indices of unitaries $U_n$ and $U_n^*$[25]. It turns out that, as in the SFF, the leading pairings are those corresponding to cyclic pairings of the set of $U_n$ and $U_n^*$. Namely, when expanding the DFF in the computational spin basis at site $n$, we have the Haar average

$$\overline{[U_n]_{a(0),a(1)} \cdots [U_n]_{a(2t-4),a(2t-3)}[U_n]_{a(2t-2),a(2t-1)}[U_n^*]_{b(0),b(1)} \cdots [U_n^*]_{b(2t-4),b(2t-3)}[U_n^*]_{b(2t-2),b(2t-1)}},$$

$$(10)$$

where we note that the phase coupling acts diagonally in this basis and that, unlike in the computation of the SFF, there is no overlap of indices as a Kraus operator is acting on the state after every unitary. Haar averaging then yields cyclic pairings of indices labeled by $a(t) = b(t+2s)$ for $s = 0, ..., t-1$ [with the periodic condition that $a(2t) = a(0)$] as the leading contributions at $q \to \infty$. See Fig. 6 for the cyclic-pairing diagrams at $t = 3$.

The difference between the SFF and the DFF, however, is that in the DFF all $s \neq 0$ pairings are suppressed compared to the $s = 0$ pairing. Indeed, the $s = 0$ pairing has weight 1, while all $s \neq 0$ pairings have the same weight smaller than 1. For instance, it is readily seen that the $s = 1$ and $s = 2$ pairings represented by Fig. 6b and 6c have the weight

$$q^{-3} \sum_{i,j,k} \text{Tr}\, M_i M_j^\dagger\, \text{Tr}\, M_j M_k^\dagger\, \text{Tr}\, M_k M_i^\dagger = \sum_{i,j,k} \eta_i \delta_{ij} \eta_j \delta_{jk} \eta_k \delta_{ki} = \sum_i \eta_i^3. \quad (11)$$

In general, at time $t$ the weight is simply given by $\kappa = \sum_{i=0}^{k-1} \eta_i^t$ for an arbitrary choice of quantum channels, and in particular, it reduces to $(1-p)^t$ when the system is dissipated by depolarizing channels. In contrast, the $s = 0$ pairing in Fig. 6a yields

$$q^{-3} \sum_{i,j,k} \text{Tr}\, M_i M_i^\dagger\, \text{Tr}\, M_j M_j^\dagger\, \text{Tr}\, M_k M_k^\dagger = \left( \sum_i \eta_i \right)^3 = 1. \quad (12)$$

Next, we move on to build the transfer matrix. This can be done by performing phase averaging on neighboring sites for every possible pairing. The situation is again the same as in the SFF in that if two sites have different pairings then the resulting weight is $e^{-\varepsilon t}$[25]. We thus have the transfer matrix of the form $T = (1 - e^{-\varepsilon t})I + e^{-\varepsilon t}E$ together with the on-site matrix $D$ that encodes different weights in $s = 0$ and $s \neq 0$ pairings. For instance, when $t = 3$ they are given by

$$T = \begin{pmatrix} 1 & e^{-3\varepsilon} & e^{-3\varepsilon} \\ e^{-3\varepsilon} & 1 & e^{-3\varepsilon} \\ e^{-3\varepsilon} & e^{-3\varepsilon} & 1 \end{pmatrix}, \quad D = \begin{pmatrix} 1 & 0 & 0 \\ 0 & \sum_{i=0}^{k-1} \eta_i^3 & 0 \\ 0 & 0 & \sum_{i=0}^{k-1} \eta_i^3 \end{pmatrix}. \quad (13)$$

With these, we obtain the exact large-$q$ DFF, $\overline{F(t)} = \text{Tr}(TD)^L = \text{Tr}\, \hat{T}^L$.

While the large-$q$ DFF we obtained in terms of the transfer matrix is exact at any $t$ and for any quantum channels and system size, it is still analytically challenging to extract any information out of it. There are two situations for which we have analytic handles, to which we turn next.

**Small system size.** The first case is when $L$ is not too large. In this case, the trace $\text{Tr}\, \hat{T}^L$ can be still computed by expressing the transfer matrix as

$$\hat{T} = \kappa T + (1 - \kappa)T|0\rangle\langle 0|. \quad (14)$$

This allows us to express the DFF as a sum of terms involving the SFF, $\overline{K(t)} = \text{Tr}\, T^L$, and the partial SFF (PSFF),

$$\overline{K_A(t)} = q^{-(L-L_A)} \overline{\text{Tr}_{\bar{A}} \left[ (\text{Tr}_A W(t)^\dagger)(\text{Tr}_A W(t)) \right]}, \quad (15)$$

where $A$ is a subsystem of size $L_A$. It is a simple matter to observe that the PSFF at large $q$ is given by $\overline{K_A(t)} = T_{00}^{L_A+1}$[34].

For instance, for $L = 3$ and $L = 4$ we have

$$\begin{aligned} \overline{F_{L=3}(t)} &= \kappa^3 \overline{K(t)} + 3\kappa^2(1-\kappa)\overline{K_2(t)} + 3\kappa(1-\kappa)^2\overline{K_1(t)} + (1-\kappa)^3\overline{F_{L=4}(t)} \\ &= \kappa^4 \overline{K(t)} + 4\kappa^3(1-\kappa)\overline{K_3(t)} + \kappa^2(1-\kappa)^2\left(4\overline{K_2(t)} + 2[\overline{K_1(t)}]^2\right) \\ &\quad + 4\kappa(1-\kappa)^3\overline{K_1(t)} + (1-\kappa)^4, \end{aligned} \quad (16)$$

but it is clear that the computation becomes intractable as $L$ increases.

**Domain-wall expansion.** Although it is always useful to have an exact closed expression, in the present work, we are mainly interested in the relaxation of the DRPM, which is characterized by the late-time behavior of the DFF. In this case, we can organize the DFF, $\overline{F(t)} = \text{Tr}\, \hat{T}^L$, which is a sum of all the pairing configurations with periodic boundary conditions, in terms of domain walls of local pairings labeled by $s = 0,\ldots, t-1$. Here what we call a domain wall is simply a configuration of two different pairings on two neighboring sites. Note that periodic boundary conditions allow only an even number of domain walls. Since each domain wall carries the entropic cost $e^{-\varepsilon t}$, the more domain walls we have in a pairing configuration, the less contribution it has at late times. This implies that the late-time behavior of the DFF is controlled by 1) pairing configurations without any domain wall and 2) those with only two domain walls.

Let us start with the first case. The absence of domain walls means that every site hosts the same pairing. Noting that while $s = 0$ pairing carries the factor 1 the other $s \neq 0$ pairings come with $\kappa$, the contributions from these configurations are $1 + (t-1)\kappa^L$ as depicted in the left panels in Fig. 7.

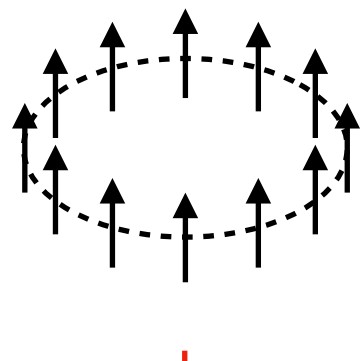

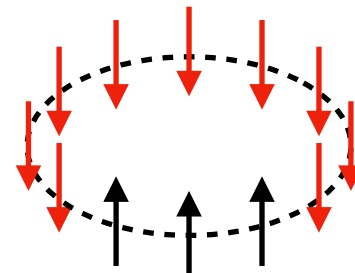

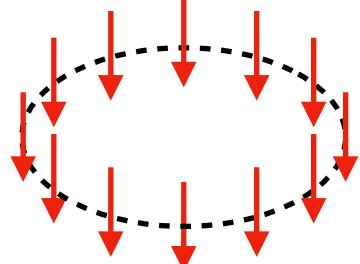

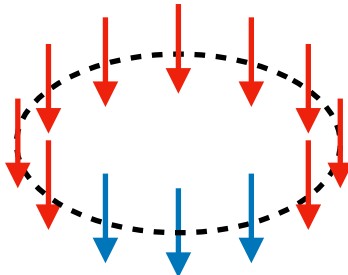

**Fig. 7 | Leading domain-wall configurations at late times.** A down spin refers to one of the $s \neq 0$ pairings (different colors mean different pairings), whereas an up spin is a $s = 0$ pairing.

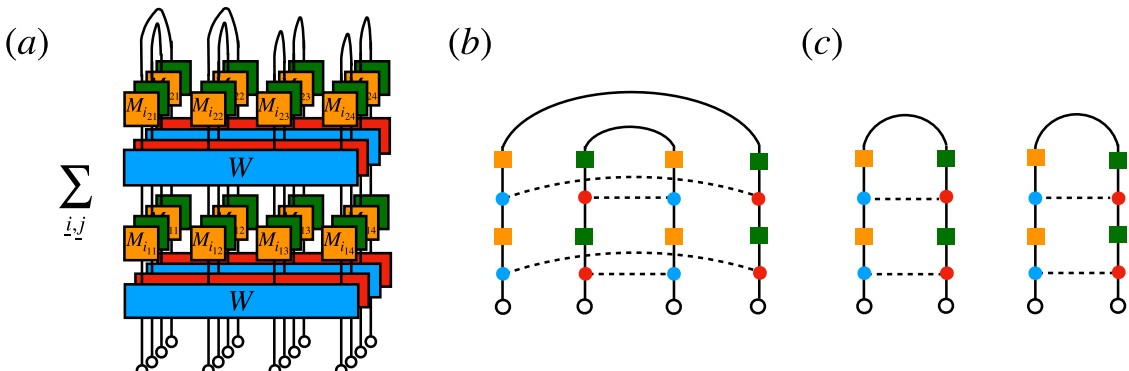

**Fig. 8 | Diagrammatic computation on the purity. a** Diagrammatic representation of the purity in the folded picture at $t = 2$ and $L_A = L_{\bar{A}} = 2$. The empty circles at the bottom represent the initial product state $|\psi\rangle$. **b, c**: Leading pairings of on-site diagrams on $A$ and $\bar{A}$ at $t = 2$, respectively.

Instead, when we have two domain walls, they divide the system into two regions, each of which is made of a fixed pairing. We can further classify them into two categories: the first type of configurations consists of one region with $s = 0$ pairings and another region with one of the $s \neq 0$ pairings (see the upper right panel in Fig. 7), and the second type of configurations is fully made of $s \neq 0$ pairings but in each region, a different one is used (e.g., $s = 1$ pairing in one region and $s = 2$ in another, see the lower right panel in Fig. 7).

Contributions from the first type can be obtained by noting that the region with $s \neq 0$ pairings could be allocated to the system in $L - a + 1$ different positions where $a$ is the size of the $s \neq 0$ pairing region, and there are $t - 1$ possible pairings. We thus have a contribution

$$(t-1)\sum_{a=1}^{L}(L-a+1)\kappa^a e^{-2\varepsilon t} = (t-1)\frac{L-(L+1)\kappa+\kappa^{L+1}}{(1-\kappa)^2}\kappa e^{-2\varepsilon t}. \quad (17)$$

Likewise, we can also evaluate the contributions from the second type, which gives

$$\frac{(t-1)(t-2)}{2}\frac{L(L-1)}{2}\kappa^L e^{-2\varepsilon t}. \quad (18)$$

Combining these, the late-time asymptotics of the DFF has the following form:

$$\overline{F(t)} \simeq 1 + t\kappa^L + tL\kappa e^{-2\varepsilon t} + \frac{t^2 L^2}{4}\kappa^L e^{-2\varepsilon t}, \quad (19)$$

where we used the fact that, at late times, $\kappa \ll 1$.

## Diagrammatic computation of the bipartite purity

While we have four replicas of $W$ and $W^\dagger$ in the bipartite purity, its computation can be carried out following the same idea as in the DFF. Again we start with representing the purity $\mathcal{P}_A(t)$ diagrammatically. In particular, we express it using the "folded" picture[24] as in Fig. 8a, where the bipartite purity at $t = 2$ in the system of size $L = 4$ with $L_A = 2$ is depicted. Note the different boundary conditions on $A$ and $\bar{A}$, which amounts to different shapes of on-site diagrams on $A$ and $\bar{A}$ as shown in Fig. 8b, c. It turns out that each diagram in Fig. 8b, c has a unique contributing pairing at large $q$, which is indicated in these diagrams. In particular, at $t = 2$, the value of the leading pairing on $A$ is

$$q^{-4}\sum_{i,j,k,l}\text{Tr}\,M_i M_j^\dagger\,\text{Tr}\,M_j M_i^\dagger\,\text{Tr}\,M_k M_l^\dagger\,\text{Tr}\,M_l M_k^\dagger = \left(\textstyle\sum_i \eta_i^2\right)^2, \quad (20)$$

whereas on $\bar{A}$ the weight is simply 1 (here, we normalized the initial product state as $|\langle\psi|\psi\rangle|^2 = 1$). For general $t$, it can be readily inferred that the weights are given by $\left(\sum_i \eta_i^2\right)^t$ and 1, respectively.

Since these two pairings are clearly different, phase averaging induces a single domain wall at the boundary of $A$ and $\bar{A}$ with the statistical cost $e^{-2\varepsilon t}$ (the factor 2 is due to four replicas of the unitaries in the purity), whereas no domain wall is produced in the bulk of $A$ and $\bar{A}$. Note that this structure is similar to what was observed in the bipartite purity of dissipationless random unitary circuits[79,80]. Combining these observations, we arrive at the ensemble-averaged purity at large $q$:

$$\overline{\mathcal{P}_A(t)} = e^{-2\varepsilon t}\left(\sum_i \eta_i^2\right)^{tL_A}. \quad (21)$$

In the case of the depolarizing channel, it reduces to

$$\overline{\mathcal{P}_A(t)} = e^{-2(1+\gamma L_A)\varepsilon t}. \quad (22)$$

## Numerical computation of the DFF and spectral gap

Numerically, the DFF of the quantum circuit can be computed by vectorizing its dynamical generator $\Phi$ and then obtaining its eigenvalues by exact diagonalization. This, however, limits the achievable system sizes considerably because of the doubling of degrees of freedom in an open quantum system. Instead, we compute the DFF by using its relation to the average survival probability of a Haar-random initial state under the evolution generated by $\mathcal{W}^{t\,57,65,81}$, which we now derive for the Kraus map of Eq. (1). To this end, we consider an initial state $\rho(0) = V\rho_R V^\dagger$, where $V$ is a $q^t \times q^t$ Haar-random unitary and $\rho_R$ is an arbitrary reference density matrix, and compute the average survival probability

$$\begin{aligned} f_{\text{surv}}(t) &= \langle\text{Tr}[\rho(t)\rho(0)]\rangle_V \\ &= \int d\mu(V)\text{Tr}\{\mathcal{W}^t[V\rho_R V^\dagger]V\rho_R V^\dagger\} \\ &= \sum_j \text{Tr}\{K_j \int d\mu(V)V\rho_R V^\dagger K_j^\dagger V\rho_R V^\dagger\}, \end{aligned} \quad (23)$$

where $d\mu(V)$ is the Haar measure over the unitary group. To proceed, we make use of the identity[82,83]

$$\begin{aligned} &\int d\mu(V)V X_1 V^\dagger X_2 V X_3 V^\dagger \\ &= \frac{D\text{Tr}[X_1 X_3] - \text{Tr}X_1\text{Tr}X_3}{D(D^2-1)}\text{Tr}X_2 I + \frac{D\text{Tr}X_1\text{Tr}X_3 - \text{Tr}[X_1 X_3]}{D(D^2-1)}X_2, \end{aligned} \quad (24)$$

for arbitrary matrices $X_{1,2,3}$ and where $D = q^t$ is the Hilbert space dimension. Using this result in Eq. (23), we find

$$f_{\text{surv}}(t) = \frac{1}{D^2-1}\left[\left(\mathcal{P}_R - \frac{1}{D}\right)F(t) + D - \mathcal{P}_R\right], \quad (25)$$

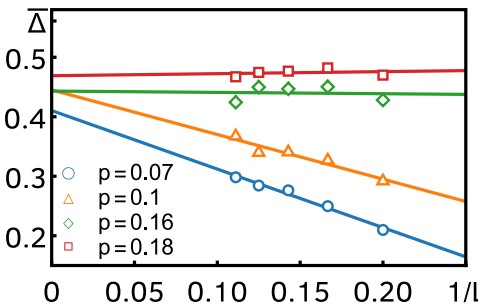

**Fig. 9 | Finite-size dependence of the spectral gap.** We plot the average spectral gap computed with the power iteration method as a function of $1/L$ for various values of $p$. The straight lines are linear fits to the data points, and the resulting intercept is the infinite-size extrapolation. In line with the existence of anomalous relaxation, for small $p$, the average gap grows with $L$, while for larger $p$ it becomes constant. In the limit $L \to \infty$, the window of $p$ with linear growth goes to 0.

where $\mathcal{P}_R = \mathrm{Tr}\rho_R^2$ is the purity of the reference state. Taking the reference state to be pure ($\mathcal{P}_R = 1$), we can write the DFF as

$$F(t) = D(D+1)f_{\mathrm{surv}}(t) - D. \tag{26}$$

To compute $f_{\mathrm{surv}}$ only matrix multiplication of operators in the original Hilbert space is done, as per Eq. (23), and no vectorization is needed. As such, we can compute $F(t)$ for systems as large as in the close case. We note that for our random circuit, an additional averaging over the Haar-random gates in each realization is performed, which in principle improves the convergence of the averaging procedure.

Finally, we consider the computation of the spectral gap. Assuming the DFF and spectral gap to be self-averaging[56,84], we can identify the decay rate of the averaged DFF,

$$\Delta = -\lim_{t\to\infty} \frac{1}{t}\log\left[\overline{F(t)} - 1\right], \tag{27}$$

with the average spectral gap $\overline{\Delta}$ of the generator $\mathcal{W}$. Numerically, the former can be computed by fitting an exponential to the late-time tail of the ensemble-averaged Eq. (26), but we find it more convenient to compute $\overline{\Delta}$, which can be done efficiently with a power-iteration method, as follows. Since the identity is the steady state of $\mathcal{W}$, applying $\mathcal{W}$ to any traceless state produces a new state that is still orthogonal to the steady state. As such, we start with a random initial state $\rho_0$ that is traceless, positive definite, and normalized as $\mathrm{Tr}\rho_0^2 = 1$. By successive application of the map $\mathcal{W}$ and normalization, $\rho_{i+1} = \mathcal{W}(\rho_i)/\sqrt{\mathrm{Tr}[\mathcal{W}(\rho_i)^2]}$, the initial state $\rho_0$ converges (up to a phase $e^{i\theta}$) to the leading non-steady state $\rho^{(1)}$, i.e., the eigenvector associated with the leading decaying eigenvalue $\lambda^{(1)}$, $\lim_{i\to\infty}\rho_i = e^{i\theta}\rho^{(1)}$ and $\mathcal{W}(\rho^{(1)}) = \lambda^{(1)}\rho^{(1)}$. At a finite iteration step $i$, $\lambda^{(1)}$ is approximated by $\lambda_i^{(1)} = \mathrm{Tr}(\rho_{i+1}\rho_i)$ and, consequently, the spectral gap is given by $\Delta = \lim_{i\to\infty}\log|\lambda_i^{(1)}|$. The deviation of $\rho_i$ from $\rho^{(1)}$ is quantified by the remainder $\epsilon_i = \sqrt{\mathrm{Tr}(r_i^\dagger r_i)} \to 0$ as $i \to \infty$, where $r_i = \rho_i - \lambda_{i-1}^{(1)}\rho_{i-1}$. In practice, we ran our algorithm until at least $\epsilon < 10^{-4}$.

To produce Fig. 5b in the Results section, we obtained $n$ values of the gap for different values of $p \in [0.01, 0.2]$ and $L = 5, \ldots, 9$. (The value of $n$ varied with $L$, ranging from $\sim 1000$ for $n = 5$ to $\sim 50$ for $L = 9$.) In Fig. 9, we plot the average gap $\overline{\Delta}$ computed for four representative values of $p$, as a function of $1/L$. The straight lines are a linear fit as a function of $1/L$, and the intercept with the $y$-axis gives our estimate for the infinite-size extrapolation [the black line in Fig. 5b].

## Data availability
The data that support the findings of this study are available from the corresponding authors upon request.

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

## Acknowledgements

T. Y. thanks Hosho Katsura for discussions out of which this work grew, and J. T. Chalker and S. J. Garratt for collaboration on a related topic. L. S. was supported by a Research Fellowship from the Royal Commission for the Exhibition of 1851 and by Fundação para Ciência e a Tecnologia (FCT-Portugal) through grant No. SFRH/BD/147477/2019. We acknowledge hospitality and support from the Simons Center for Geometry and Physics and the program "Fluctuations, Entanglements, and Chaos: Exact Results", where this work was initiated. This project was partially funded by the QuantERA II Program and has received funding from the European Union's Horizon 2020 research and innovation program under Grant Agreement No. 101017733 (https://doi.org/10.54499/QuantERA/0003/2021).

## Author contributions

T. Y. and L. S. contributed equally to all aspects of this work.

## Competing interests

The authors declare no competing interests.
