## [Transparent Peer Review file · Nature Communications]

Robustness of Quantum Chaos and Anomalous Relaxation in Open Quantum Circuits

Corresponding Author: Dr Lucas Sá

Version 1:

Reviewer comments:

Reviewer #1

(Remarks to the Author)

The authors studied the dissipative form factor (DFF) using the dissipative random phase model and the dissipative brickwork model. They identified various relaxation behaviour of DFF as a function of the dependence of dissipation strength on the system size. They also argued for the existence of anomalous relaxation, which refers to the existence of a finite size gap if the thermodynamic limit is taken before the dissipationless limit is taken. The authors have two sets of results. Analytically, in large time t , they derived Eq. 3 when the size of the on-site Hilbert space dimension q is taken to infinity. Numerically, they have simulated the brickwork model at finite q .

On the analytical derivation, it is unclear to me whether this derivation applies only the depolarizing channel. Above Eq. 10, the authors have identified a set of diagrams that dominate. Does the derivation require depolarizing channel (currently "depolarizing channel" is only mentioned once in the appendices it seems)? Why do only these diagrams apply for generic Kraus operator? Similarly for the calculation with bipartite purity. Why only these diagrams, and is this depolarizing channel only?

I am skeptical on the importance of this work. I find the set-up to be quite unphysical and somewhat artificial. The figure for the dynamical regimes, figure 3, is based on the fact that the dissipation strength is vanishing in large L . As the authors themselves noted, the most natural choice is $\alpha = 0$. The authors stated that "rescaling of the local dissipation strength with system size is in some cases necessary to get a well-defined theory in the thermodynamic limit," but I don't believe this statement suffice in giving physical motivation to this set-up.

The authors used the phrase "universal" multiple times in the manuscript. While the spectral form factor is a good probe of quantum chaos, DFF behavior is dominated by the gap of the spectrum. As far as I know, spectral gap is not known to be "universal". Which aspects of the results are "universal"? What does it mean to be "universal" here, beyond the spectral correlation? Can the authors simulate a second models and show that the same behaviour is displayed, e.g. Fig 3? Since figure 5 is supposed to be main support of the existence of anomalous relaxation at finite values of q . I think the authors should provide quite a bit more details on this, e.g. the finite-size scaling plots. I am hesitant to call the figure 5 "smoking-gun signature of anomalous relaxation" at the moment.

For readability, it may be good to have all the scaling behavior of t_* , t_{Th} , t_d , and t_{DW} to be at the same place (for example, in a table).

Due to the above issues, I do not recommend the manuscript to be published in Nature Communication. It should be published in a more specialised journal when the above issues can be clarified.

Reviewer #2

(Remarks to the Author)

In the manuscript, the authors investigated the dissipative form factor (DFF) in the dissipative random phase model, which is analytically computable in the limit of large local Hilbert-space dimension. They found two different regimes: in the regime where the dissipation strength scales with the system size L as L^a with $a < -1$, the "ramp" behavior of the DFF is observed for a long time interval, whereas in the regime where $a > -1$, the exponential decay without the ramp is observed. Interestingly, in

the latter regime, as long as we take the thermodynamic limit first, we have a finite decay rate even in the limit of the weak dissipation, which is dubbed anomalous relaxation.

I think that this manuscript reports novel results based on solid theoretical calculations. Although interesting "anomalous relaxation" phenomena have already been known and studied in recent literatures, I think that it is valuable to propose an exactly solvable model exhibiting such phenomena. The authors also carried out numerical calculations on another model (the brickwork random Floquet circuit), which clearly show that the behavior observed in the exactly solvable model is not due to the peculiarity of the model.

However, I find some issues listed below, so the authors should fix them before I can recommend publication.

(1) Comparing Fig. 2 (a) and (b), we see a qualitative difference. In Fig. 2(a), there is a dip before the ramp, but there is no dip in Fig. 2(b). This might not be of great importance, but I would like to ask the authors to briefly explain the origin of this difference.

(2) At line 256, Figure 3 was referred to in the main text, but at that stage the parameter γ is not defined (γ appears at line 273). So I feel that it would be helpful to give an explicit definition of γ in the caption.

(3) At line 415, the authors say "the bipartite purity is expected to capture the time evolution of the strength of entanglement between A and \bar{A} ", but it is correct only for pure states. After dissipative time evolution, the quantum state of the total system is described by a mixed state, and the subsystem purity is not a measure of the entanglement between two subsystems. The authors should fix it.

(4) At line 439, $\text{Tr}[\rho(t)]^2$ should be $\text{Tr}[\rho(t)^2]$.

(5) Based on Eq. (8), the authors conclude that anomalous relaxation is absent from the purity dynamics, but I do not think that such a strong conclusion is drawn from Eq. (8). My concern is that the purity $P_A(t)$ is not a self-averaging quantity: even if its ensemble average ("annealed average") does not show anomalous relaxation, it does not exclude the occurrence of anomalous relaxation in an individual sample. Instead, the logarithm $\log[P_A(t)]$ would be self-averaging, so one could draw a general conclusion by computing the ensemble average of $\log[P_A(t)]$ ("quenched average").

(6) At line 455, the authors say "Interestingly, one signature of quantum chaos in the spectral form factor, i.e., the late-time ramp, is robust against weak dissipation", but I feel uneasy about this statement. The late-time ramp is observed only when the dissipation strength is smaller than $(L \log L)^{-1}$, which is extremely small in a many-body system. As the authors mentioned, a local dissipation strength that is independent of the spatial extent of the system is the most natural choice. There is no ramp in such a natural choice. Of course, in a realistic experimental situation, the condition $\gamma < (L \log L)^{-1}$ would be satisfied. However, I feel that it is rather trivial that the ramp behavior is somehow robust against adding weak dissipation. In other words, for any finite-size system, it is rather trivial that the ramp behavior is observed for sufficiently small γ . What is non-trivial here is that the threshold is proportional to $(L \log L)^{-1}$. I feel that the authors should weaken the statement around lines 455-457.

Reviewer #3

(Remarks to the Author)

Summary: In this manuscript, the authors study the Dissipative Random Phase model (DRPM) as a paradigmatic model of dissipative Floquet dynamics. This model is proposed in the manuscript as a dissipative extension of the Random Phase Model which has been extensively studied. To study the dynamics, the authors primarily study a Dissipative Spectral Form Factor (DFF), which they evaluate analytically in the limit of large local Hilbert space dimension. The analytical results are corroborated with numerical computation of finite dimensional brickwork circuits. The authors also compute the purity as a probe of entanglement dynamics in this setup.

The main findings of this manuscript center around two limits: the weak dissipation limit $\gamma \rightarrow 0$ and the thermodynamic limit $L \rightarrow \infty$. The authors demonstrate that the DFF shows starkly distinct behavior depending on the order of these limits. Additionally, in the presence of dissipation the DFF is shown to possess two peaks: a Thouless peak (which also exists in the absence of dissipation) and a dissipation peak arises due to competition between suppression due to dissipation and the (universal) linear ramp. The decay after the dissipation peak is termed as a spectral gap Δ_d which is shown to persist for small dissipation if the thermodynamic limit is taken first. This is the phenomenon of anomalous relaxation (which has been observed in other non-Floquet dissipative systems). For dissipation strength that decays sufficiently strongly with system size, the ramp of the DFF is found to be robust. The authors also introduce the notion of domain walls and quantify their contribution to the dynamics, as well as the timescales in which these become relevant.

The authors propose that the observed behavior is generic to open quantum circuits.

Recommendation: The results obtained by the authors are novel and interesting. The conclusions and claims are well supported and thorough. The questions answered in the manuscript are of a general interest to the large community of

researchers working in this and related fields. I am happy to recommend the manuscript for publication.

Comments/Questions for the authors:

1. The authors perform an averaging procedure over Haar-unitaries in order to determine the averaged DFF. It is known that averaging is equivalent to an additional non-unitary channel. Can the authors comment on the relevance of this unitary channel to their results?
2. A relevant parameter that the authors use is $\kappa := \sum_i \eta^t_i$, which they argue to be $\ll 1$ for large t . This is seemingly only true if $\eta_i < 1$, which is not emphasised explicitly in the main text. Is it somehow implicit in the formalism? If not, then do the authors foresee any interesting consequences of $\eta_i > 1$?
3. The ramp of the DFF seems to persist for small p while it disappears for large p . Is there a transition point/cutoff dissipation p_0 at which the ramp vanishes?
4. In the Brickwork numerics, while the behavior is qualitatively similar to the analytical DRPM, there seems to be some quantitative distinctions. Specifically the ramp seems to disappear at much smaller p in the Brickwork circuit. Is it purely a finite size effect? Or does it correspond to some finer difference?
5. The Thouless peak is a signature of early time physics, while the dissipation peak emerges at late times. It is demonstrated that the Thouless peak is controlled by the domain wall contribution F_{DW} . However, from the discussion in the Methods supplementary, it appears as if the domain wall contributions (esp. $\neq 0$) emerge at late times. The authors are requested to clarify this potentially confusion in the main text.

Version 2:

Reviewer comments:

Reviewer #1

(Remarks to the Author)

I find that the authors have adequately answered my questions and addressed my concerns. Given the other two referees find the work novel, I am happy to recommend this manuscript for publication at Nature Communication.

Reviewer #2

(Remarks to the Author)

I read authors' reply and the revised manuscript. I have confirmed that my concerns are fully addressed. Although the dissipative random phase model is somewhat artificial, it is of great value that the anomalous relaxation is exactly derived. I'm therefore happy to recommend the publication of this manuscript in Nature Communications.

Reviewer #3

(Remarks to the Author)

I thank the authors for answering my questions. I am satisfied with the explanation and am happy to recommend the article for publication.

Response to the Report of Reviewer #1

We thank Referee 1 for carefully reading our paper and for their detailed comments, which we now address.

(We reproduce the referee’s comments in gray and give our answer below each comment.)

The authors studied the dissipative form factor (DFF) using the dissipative random phase model and the dissipative brickwork model. They identified various relaxation behaviour of DFF as a function of the dependence of dissipation strength on the system size. They also argued for the existence of anomalous relaxation, which refers to the existence of a finite size gap if the thermodynamic limit is taken before the dissipationless limit is taken. The authors have two sets of results. Analytically, in large time t , they derived Eq. 3 when the size of the on-site Hilbert space dimension q is taken to infinity. Numerically, they have simulated the brickwork model at finite q .

On the analytical derivation, it is unclear to me whether this derivation applies only the depolarizing channel. Above Eq. 10, the authors have identified a set of diagrams that dominate. Does the derivation require depolarizing channel (currently “depolarizing channel” is only mentioned once in the appendices it seems)? Why do only these diagrams apply for generic Kraus operator? Similarly for the calculation with bipartite purity. Why only these diagrams, and is this depolarizing channel only?

We thank the referee for pointing out that this issue was not sufficiently clear in the manuscript. The diagrammatic derivation we perform in the Methods section applies to *any* choice of quantum channels that act on each site independently (generalizations to quantum channels acting on multiple sites are straightforward). Only diagrams associated with cyclic pairings of Haar unitaries contribute at large q upon Haar averaging because all other diagrams carry higher powers of q^{-1} , which are suppressed in the large- q limit. The mechanism is exactly the same as in the SFF of the RPM, where t cyclic pairings with the *same* weight give rise to the late-time ramp $\overline{K(t)} \simeq t$ (in the DFF of the DRPM, the weight of $s = 0$ pairing is *different* from that of $s \neq 0$ pairings and, therefore, the DFF shows more intricate behaviors). See arXiv:2312.14234 for extensive discussions on how to single out contributing pairings in different Haar-averaged quantities in the RPM. Similarly, we can also identify the most relevant pairings in the purity. In this case, the situation is simpler than in the DFF as only two pairings, see Fig. 8(b,c), have the leading contributions at large q .

In Methods, we focus on the case where the same quantum channel acts on every site, but the generalization to the case where each site is dissipated by a different quantum channel is trivial and given in the main text. This also implies that the exact DFF at large q , Eq. (2), as well as its domain-wall expansion, Eq. (3), are valid for any quantum channels.

Choosing a particular quantum channel is, however, necessary to study the behavior of the domain-wall expansion quantitatively, and for that, we choose to work with depolarizing channels after Eq. (3). To make it clear that all the results up to Eq. (3) are valid for any quantum channel, we added a sentence after Eq. (3) that reads “It should be stressed that the late-time expansion Eq. (3) is still applicable to any quantum channel.”

I am skeptical on the importance of this work. I find the set-up to be quite unphysical and somewhat artificial. The figure for the dynamical regimes, figure 3, is based on the fact that the dissipation strength is vanishing in large L . As the authors themselves noted, the

most natural choice is $\alpha = 0$. The authors stated that “rescaling of the local dissipation strength with system size is in some cases necessary to get a well-defined theory in the thermodynamic limit,” but I don’t believe this statement suffice in giving physical motivation to this set-up.

The main idea behind the DRPM is to introduce an open quantum many-body system that is as simple as possible yet still physical by stripping out unnecessary properties such as continuous time-translation symmetry and conservation laws, which are retained in more physical systems, e.g. Hamiltonian systems. The practice of studying such minimally structured models has been rather prevalent in the study of quantum chaotic many-body dynamics in recent years. The exact results obtained in these studies thanks to the simple structure of the systems have offered us considerable insights on the nature of *isolated* quantum chaotic many-body systems (see e.g. [Phys. Rev. X **8**, 021014 (2018)] and [Phys. Rev. Lett. **121**, 060601 (2018)]). Here we wish to do the same for *open* quantum many-body systems and, to this end, we proposed the DRPM as an ideal playground to investigate the universal properties in chaotic open quantum many-body systems. We choose an open Floquet system to study the interplay between quantum chaos and dissipation without having to worry about any influence of conservation laws. We also emphasize that Floquet circuits dissipated by quantum channels are in fact physically realistic models for NISQ computers.

Regarding the assumed scaling form of the effective dissipation strength γ , we reiterate that in many-body systems it might be necessary to rescale the dissipation strength so as to avoid divergence in the thermodynamic limit (similarly to the rescaling of interactions to bring it to the same scaling of the kinetic term in Hamiltonian systems). This is a common practice, e.g., in the literature on the Sachdev-Ye-Kitaev model (both closed and open). In any case, we note that the scaling form does not exclude the choice of α that makes γ constant ($\alpha = 0$) or even divergent ($\alpha > 0$). In fact, we have studied the behavior of the DFF in all possible regimes, and our most striking result—the analytical proof of anomalous relaxation—is valid in the regime that the referee considers “physical”.

The authors used the phrase “universal” multiple times in the manuscript. While the spectral form factor is a good probe of quantum chaos, DFF behavior is dominated by the gap of the spectrum. As far as I know, spectral gap is not known to be “universal”. Which aspects of the results are “universal”? What does it mean to be “universal” here, beyond the spectral correlation? Can the authors simulate a second models and show that the same behaviour is displayed, e.g. Fig 3?

We thank the referee for raising this important question. First, we believe that the main features of the DFF in the DRPM that we identified in the paper, namely, the robustness of the ramp, the competition between the Thouless and dissipation peaks, and the presence of anomalous relaxation, are all “universal” in one-dimensional open quantum many-body systems without conservation laws, provided that dissipation is introduced in the same way as in the DRPM. This is because the SFF of the RPM, which is an archetypal *isolated* spatially-extended quantum chaotic system without conservation laws, is entirely controlled by the domain-wall physics, and it is also known that a similar effective domain-wall structure always governs the behavior of the SFF in a generic quantum many-body system without a conserved charge even at finite q [Phys. Rev. X, **11**, 021051 (2021)]. Since all the above characteristic behaviors of the DFF are again induced by the domain-wall physics in the DRPM, it is natural to assume that the same mechanism is also at play in generic open quantum many-body systems without conservation laws, and in this sense they are universal.

We add that anomalous relaxation, in particular, has been recently proposed as a so-far unexplored

universal signature of many-body quantum chaos (dissipative or not), see, [Phys. Rev. D **107**, 106006 (2023)] and [Phys. Rev. B **109**, 064311 (2024)]. That is, the *value* of the gap is not universal, but the fact that it exhibits this noncommutativity of limits when the system is chaotic, rendering it very large even when the dissipation is infinitesimally weak, is.

In light of the above discussion, we expect that similar relaxation behaviors as in Fig. 3 would be observed for any Floquet circuits (note that Fig. 3 is obtained analytically and we cannot simulate it for another model). To support this claim, in Fig. 2 we provided numerical evidence for the brickwork Floquet circuit with $q = 2$, which qualitatively shows the behaviors described above.

Since figure 5 is supposed to be main support of the existence of anomalous relaxation at finite values of q . I think the authors should provide quite a bit more details on this, e.g. the finite-size scaling plots. I am hesitant to call the figure 5 “smoking-gun signature of anomalous relaxation” at the moment.

From our analytical results on the DRPM, we know that the universal features—in particular the simultaneous presence of both Thouless and dissipation peaks—only become completely salient for L larger (or even much larger) than those attainable with today’s numerics. Anomalous relaxation is the only feature that we can study quantitatively for the available system sizes. Nevertheless, this is still with its challenges, since we have to take the $L \rightarrow \infty$ limit before the $\gamma \rightarrow 0$ when we only have access to $L \lesssim 10$. For that reason, we argue that the order-one infinite-size extrapolation from the data down to small but finite p , together with the increasing slope at the origin is, indeed, the smoking-gun signature of anomalous relaxation. To convince the referee of this point, in the revised version, we added a new panel to Fig. 5, where we plot the *exact* finite- L gap for the DRPM, Eq. (7), for the same small values L . At small L , we can see that the plot is qualitatively the same as for the brickwork numerics and thus we claim that these finite- L features are, indeed, smoking-gun signatures of anomalous relaxation. We have emphasized this point in the updated manuscript. As requested by the referee, we have also added the finite-size scaling analysis, for some representative values of p , to the Methods section.

In light of the above discussion on the difficulty of numerically studying dissipative many-body quantum chaos at finite L , we hope the referee agrees that our exact analytical results assume an even more important role deserving broad dissemination.

For readability, it may be good to have all the scaling behavior of t_ , t_{Th} , t_d , and t_{DW} to be at the same place (for example, in a table).*

We are grateful for this useful suggestion—we added a table in the new version.

Due to the above issues, I do not recommend the manuscript to be published in Nature Communication. It should be published in a more specialised journal when the above issues can be clarified.

We again thank the referee for their valuable comments. We believe that we fully addressed the questions and comments of the referee, provided clarifications, and implemented the necessary changes in the manuscript, which greatly increased readability. We hope that the referee will now consider our work to meet the high standards of Nature Communications.

Response to the Report of Reviewer #2

We thank Referee 2 for carefully reading our paper and for their detailed comments, which we now address.

(We reproduce the referee’s comments in gray and give our answer below each comment.)

In the manuscript, the authors investigated the dissipative form factor (DFF) in the dissipative random phase model, which is analytically computable in the limit of large local Hilbert-space dimension. They found two different regimes: in the regime where the dissipation strength scales with the system size L as L^a with $a < -1$, the “ramp” behavior of the DFF is observed for a long time interval, whereas in the regime where $a > -1$, the exponential decay without the ramp is observed. Interestingly, in the latter regime, as long as we take the thermodynamic limit first, we have a finite decay rate even in the limit of the weak dissipation, which is dubbed anomalous relaxation.

I think that this manuscript reports novel results based on solid theoretical calculations. Although interesting “anomalous relaxation” phenomena have already been known and studied in recent literatures, I think that it is valuable to propose an exactly solvable model exhibiting such phenomena. The authors also carried out numerical calculations on another model (the brickwork random Floquet circuit), which clearly show that the behavior observed in the exactly solvable model is not due to the peculiarity of the model.

We thank the referee for their summary and overall very positive assessment of our work.

(1) Comparing Fig. 2 (a) and (b), we see a qualitative difference. In Fig. 2(a), there is a dip before the ramp, but there is no dip in Fig. 2(b). This might not be of great importance, but I would like to ask the authors to briefly explain the origin of this difference.

This is purely a finite-size effect. As commented in the manuscript, for the available system sizes, the Thouless peak is not visible, see also Fig. 5 of [Phys. Rev. X, **11**, 021051 (2021)]. (Note that because our model is nonunitary, there is either a doubling of degrees of freedom or a sum over quantum trajectories, which takes the larger sizes studied in that paper out of our reach.) To illustrate this point, we computed the exact DFF for the DRPM at the same small $L = 8$ (for comparison, we reproduce Fig. 1b):

As can be seen, the results are qualitatively the same. In the paper, we opted to include the plot of the DRPM at $L = 12$ as it better illustrates our main analytical findings.

(2) At line 256, Figure 3 was referred to in the main text, but at that stage the parameter γ is not defined (gamma appears at line 273). So I feel that it would be helpful to give an explicit definition of gamma in the caption.

We have followed the referee’s suggestion.

(3) At line 415, the authors say “the bipartite purity is expected to capture the time evolution of the strength of entanglement between A and \bar{A} ”, but it is correct only for pure states. After dissipative time evolution, the quantum state of the total system is described by a mixed state, and the subsystem purity is not a measure of the entanglement between two subsystems. The authors should fix it.

We agree with the referee and have removed this sentence.

(4) At line 439, $\text{Tr}[\rho(t)]^2$ should be $\text{Tr}[\rho(t)^2]$.

We have fixed this typo.

(5) Based on Eq. (8), the authors conclude that anomalous relaxation is absent from the purity dynamics, but I do not think that such a strong conclusion is drawn from Eq. (8). My concern is that the purity $P_A(t)$ is not a self-averaging quantity: even if its ensemble average (“annealed average”) does not show anomalous relaxation, it does not exclude the occurrence of anomalous relaxation in an individual sample. Instead, the logarithm $\log[P_A(t)]$ would be self-averaging, so one could draw a general conclusion by computing the ensemble average of $\log[P_A(t)]$ (“quenched average”).

The referee makes an interesting suggestion. Unfortunately, using our diagrammatic techniques we cannot analytically compute the quenched average $\exp(-S_A^{(2)}(t))$. We have mentioned this possibility in the text and weakened the claim of absence of anomalous relaxation.

(6) At line 455, the authors say “Interestingly, one signature of quantum chaos in the spectral form factor, i.e., the late-time ramp, is robust against weak dissipation”, but I feel uneasy about this statement. The late-time ramp is observed only when the dissipation strength is smaller than

$$(L \log L)^{-1}$$

, which is extremely small in a many-body system. As the authors mentioned, a local dissipation strength that is independent of the spatial extent of the system is the most natural choice. There is no ramp in such a natural choice. Of course, in a realistic experimental situation, the condition $\gamma < (L \log L)^{-1}$ would be satisfied. However, I feel that it is rather trivial that the ramp behavior is somehow robust against adding weak dissipation. In other words, for any finite-size system, it is rather trivial that the ramp behavior is observed for sufficiently small gamma. What is non-trivial here is that the threshold is proportional to $(L \log L)^{-1}$. I feel that the authors should weaken the statement around lines 455-457.

We fully agree with this comment and accordingly modified the statement, which now reads “We also observed that one signature of quantum chaos in the spectral form factor, i.e., the late-time ramp, is robust when dissipation is sufficiently weak, $\gamma \lesssim (L \log L)^{-1}$, and its remnant persists over a timescale controlled by the gap.”

However, I find some issues listed below, so the authors should fix them before I can recommend publication.

We again thank Referee 2 for their thorough reading of our paper and their many useful comments. We hope that, with the clarifications provided above and the changes effected to the manuscript, they will now consider our work to meet the high standards of Nature Communications.

Response to the Report of Reviewer #3

We thank Referee 3 for carefully reading our paper and for their detailed comments, which we now address.

(We reproduce the referee's comments in gray and give our answer below each comment.)

Summary: In this manuscript, the authors study the Dissipative Random Phase model (DRPM) as a paradigmatic model of dissipative Floquet dynamics. This model is proposed in the manuscript as a dissipative extension of the Random Phase Model which has been extensively studied. To study the dynamics, the authors primarily study a Dissipative Spectral Form Factor (DFF), which they evaluate analytically in the limit of large local Hilbert space dimension. The analytical results are corroborated with numerical computation of finite dimensional brickwork circuits. The authors also compute the purity as a probe of entanglement dynamics in this setup.

The main findings of this manuscript center around two limits: the weak dissipation limit $\gamma \rightarrow 0$ and the thermodynamic limit $L \rightarrow \infty$. The authors demonstrate that the DFF shows starkly distinct behavior depending on the order of these limits. Additionally, in the presence of dissipation the DFF is shown to possess two peaks: a Thouless peak (which also exists in the absence of dissipation) and a dissipation peak arises due to competition between suppression due to dissipation and the (universal) linear ramp. The decay after the dissipation peak is termed as a spectral gap Δ_d which is shown to persist for small dissipation if the thermodynamic limit is taken first. This is the phenomenon of anomalous relaxation (which has been observed in other non-Floquet dissipative systems). For dissipation strength that decays sufficiently strongly with system size, the ramp of the DFF is found to be robust. The authors also introduce the notion of domain walls and quantify their contribution to the dynamics, as well as the timescales in which these become relevant.

The authors propose that the observed behavior is generic to open quantum circuits.

We thoroughly agree with the referee's summary and thank them for their careful reading.

The results obtained by the authors are novel and interesting. The conclusions and claims are well supported and thorough. The questions answered in the manuscript are of a general interest to the large community of researchers working in this and related fields. The manuscript is suited for publication. (\dots) I would be happy to recommend the manuscript for publication.

We are grateful for the very positive assessment and recommendation by the referee for publication.

Comments/Questions for the authors:

- 1. The authors perform an averaging procedure over Haar-unitaries in order to determine*

the averaged DFF. It is known that averaging is equivalent to an additional non-unitary channel. Can the authors comment on the relevance of this unitary channel to their results?

The Haar-averaging by itself does not introduce any nonunitarity. The same averaging is performed in the standard RPM, whose dynamics is fully unitary, and it leads to the usual SFF with a linear ramp and no decay. The nonunitarity of the DRPM arises only because of the explicit insertion of single-site quantum channels.

2. A relevant parameter that the authors use is $\kappa := \sum_i \eta_i^t$, which they argue to be $\ll 1$ for large t . This is seemingly only true if $\eta_i < 1$, which is not emphasised explicitly in the main text. Is it somehow implicit in the formalism? If not, then do the authors foresee any interesting consequences of $\eta_i > 1$?

Indeed, we always have $\eta_i < 1$. This can be seen from the trace-preservation of the channel: tracing the sum rule $\sum_i M_i^\dagger M_i = I$ and using $\text{Tr}(M_i M_j^\dagger) = q \eta_i \delta_{ij}$, we obtain $\sum_i \eta_i = 1$, whence it follows that $\eta_i \leq 1$ (they must be positive because of the complete positivity of the map). We have made this requirement explicit in the text.

3. The ramp of the DFF seems to persist for small p while it disappears for large p . Is there a transition point/cutoff dissipation p_0 at which the ramp vanishes?

As discussed in the paper, the ramp ceases to exist if $\gamma \gtrsim (L \log L)^{-1}$. This is not a sharp transition at a critical γ but rather a crossover when γ grows faster than a certain inverse power of L .

4. In the Brickwork numerics, while the behavior is qualitatively similar to the analytical DRPM, there seems to be some quantitative distinctions. Specifically the ramp seems to disappear at much smaller p in the Brickwork circuit. Is it purely a finite size effect? Or does it correspond to some finer difference?

As the referee points out rightly, this is indeed purely a finite-size effect. As commented in the manuscript, for the available system sizes, the Thouless peak is not visible, see also Fig. 5 of [Phys. Rev. X, **11**, 021051 (2021)]. (Note that because our model is nonunitary, there is either a doubling of degrees of freedom or a sum over quantum trajectories, which takes the larger sizes studied in that paper out of our reach.) To illustrate this point, we computed the exact DFF for the DRPM at the same small $L = 8$ (for comparison, we reproduce the Fig. 1b):

As can be seen, the results are qualitatively the same. In the paper, we opted to include the plot of

the DRPM at $L = 12$ as it better illustrates our main analytical findings.

5. The Thouless peak is a signature of early time physics, while the dissipation peak emerges at late times. It is demonstrated that the Thouless peak is controlled by the domain wall contribution F_{DW} . However, from the discussion in the Methods supplementary, it appears as if the domain wall contributions (esp. $s \neq 0$) emerge at late times. The authors are requested to clarify this potentially confusion in the main text.

The Thouless peak is a signature of early times only if compared to the Heisenberg time. However, since $t_{\text{Th}} \sim \log L$ for our purposes one should consider it as a late-time phenomenon when L is sufficiently large. As we write in the main text, the domain-wall expansion is valid around the Thouless time, which is the physical timescale over which the system as a whole starts acting on the entire system.

Recommendation: The results obtained by the authors are novel and interesting. The conclusions and claims are well supported and thorough. The questions answered in the manuscript are of a general interest to the large community of researchers working in this and related fields. I am happy to recommend the manuscript for publication.

We again thank Referee 3 for their thorough reading of our paper and their useful comments. Referee 3 already recommended the manuscript for publication and we have addressed all their queries.